# Complete Androgen Insensitivity Syndrome in a Young Girl with Primary Amenorrhea and Suspected Delayed Puberty: A Case-Based Review of Clinical Management, Surgical Follow-Up, and Oncological Risk

**DOI:** 10.3390/diseases12100235

**Published:** 2024-10-01

**Authors:** Barbara Fraccascia, Giorgio Sodero, Lucia Celeste Pane, Elena Malavolta, Caterina Gola, Luigi Pane, Valentina Filomena Paradiso, Lorenzo Nanni, Donato Rigante, Clelia Cipolla

**Affiliations:** 1Department of Life Sciences and Public Health, Fondazione Policlinico Universitario A. Gemelli IRCCS, 00168 Rome, Italy; barbara.fraccascia01@icatt.it (B.F.); elena.malavolta01@icatt.it (E.M.); caterina.gola01@icatt.it (C.G.); donato.rigante@unicatt.it (D.R.); clelia.cipolla@policlinicogemelli.it (C.C.); 2Dipartimento della Donna, del Bambino e di Chirurgia Generale e Specialistica, Università della Campania Luigi Vanvitelli, 81100 Naples, Italy; luigi.pane@studenti.unicampania.it; 3Unit of Pediatric Surgery, Fondazione Policlinico Universitario A. Gemelli IRCCS, 00168 Rome, Italy; filomenavalentina.paradiso@unicatt.it (V.F.P.); lorenzo.nanni@policlinicogemelli.it (L.N.); 4Università Cattolica Sacro Cuore di Roma, 00168 Rome, Italy

**Keywords:** complete androgen insensitivity syndrome, sexual development, disorders of sex development, pediatric endocrinology, personalized medicine

## Abstract

*Background:* Complete androgen insensitivity syndrome (CAIS) is a rare disorder of sex development characterized by 46,XY karyotype and testes, yet presenting with a complete female phenotype, which is related to mutations in the androgen receptor (*AR*) gene. *Case presentation:* We herein present the case of a 14-year-old adolescent with primary amenorrhea and suspected delayed puberty whose diagnostic journey led to the identification of CAIS through the demonstration of a novel *AR* variant (c.159_207del). *Case-based review:* Our report encompasses the complexity of CAIS management, focusing on the risk of malignancy, surveillance options, hormone replacement therapy, timing of an eventual gonadectomy, and the psychosocial impact of such a diagnosis. An algorithm has been formulated for the management of CAIS starting in adolescence, highlighting the conservative approach for those patients unwilling to undergo gonadectomy. *Conclusions:* Primary amenorrhea and delay in puberty development may provide clues, ultimately leading to a diagnosis of CAIS. This review emphasizes the cruciality of a multidisciplinary approach in managing patients with CAIS, needing for an individualized care to optimize the overall outcome.

## 1. Introduction

Disorders of sex development (DSD) include a group of congenital conditions in which the development of chromosomal, gonadal, or anatomical sex is atypical: they may present with ambiguity in the external genitalia or discordance between chromosomal, gonadal, and phenotypic sex characteristics [1]. In case of discordance between phenotype and genotype in a newborn (with male chromosomal sex and female external genitals), among the various recommended investigations, genetic resistance to testosterone should be first ruled out [2]. Complete androgen insensitivity syndrome (CAIS), also named Morris syndrome, is a rare disorder of sex development classified by the National Institutes of Health’s Rare Diseases as a condition affecting approximately 1 in 20,000–99,000 XY individuals, resulting in a complete female phenotype, despite having a 46,XY karyotype and testes as gonads. The primary cause of CAIS is related to mutations in the androgen receptor (*AR*) gene, which is encoded by a single-copy gene at the Xq11–12 locus, and diagnosis can be accordingly confirmed by *AR* sequencing on the X chromosome [3]. Clinically, individuals with CAIS present with female phenotype: during the neonatal period, they could exhibit female external genitalia and inguinal palpable gonads; during puberty, this condition could cause primary amenorrhea and a lack of axillary or pubic hair; biochemically, they may also exhibit elevated testosterone levels combined with regular 46,XY karyotype [4]. The management of CAIS may require gonadectomy although the decision-making process is troubling and requires extreme caution. Our case highlights the diagnostic process of a young adolescent presenting with primary amenorrhea and suspected delayed puberty, ultimately leading to a diagnosis of CAIS confirmed by a novel *AR* variant.

## 2. Case Presentation

A 14-year-old girl, phenotypically presenting as a female, sought medical attention due to amenorrhea; her physical examination revealed female external genitalia with Tanner stage I breast development and absence of both axillary and pubic hair. The patient’s height fell within the height target range (target height of 168 cm); furthermore, her growth rate was within normal limits for her age.

The girl underwent pelvic ultrasound evaluation, showing a blind-ended vaginal canal, absent uterus, left gonad localized in an extra-pelvic location (near the rectum), enlarged due to the presence of an unilocular cyst with anechoic content (measuring 47 × 49 × 48 mm), non-vascularized on the power Doppler; the right gonad was also visualized in extra-pelvic location (lower abdomen), with regular dimensions and morphology (16 × 11 × 17 mm). Abdominal magnetic resonance imaging (MRI) revealed a residue of gonadal tissue (a cystic formation of approximately 3.5 cm in the left adnexal region); cholelithiasis was also documented. Our patient’s hormonal assessment showed elevated testosterone levels, along with normal levels of luteinizing hormone (LH) and follicle-stimulating hormone (FSH), while estradiol was within the male reference range. The anti-Müllerian hormone (AMH) was 12 ng/mL, while the level of inhibin B was 54 pg/mL. Both values were within the normal range for her age, with the hormonal cut-offs used in our center being comparable to those previously reported [5].

The karyotype revealed a 46,XY configuration (standard human karyotype analysis, at least 20 metaphases examined). Molecular analysis of the *AR* gene, utilizing genomic DNA extracted from peripheral blood cells, revealed hemizygosity for the c.159_207del mutation (p.Leu54Serfs*105); this *AR* variant was identified using PCR followed by Sanger sequencing conducted in accordance with the Recommendations of the American College of Medical Genetics and Genomics and the Association for Molecular Pathology [6]. The identified mutation is not described in the medical literature and has been interpreted as “likely pathogenic” because it induces the formation of a premature stop codon, altering *AR* transcription and resulting in the production of a truncated and nonfunctional protein; subsequent genetic evaluation confirmed the suspicion of CAIS. Therefore, our clinical diagnosis was accordingly confirmed.

Given the oncological risk involving the gonadal residue, the patient was admitted to the division of Pediatric Surgery at 15 years for laparotomy and potential gonadectomy. During exploration, the left gonad exhibited a large cyst and two hydatids, while the right one had regular dimensions with a small hydatid: the hydatid of the right gonad was removed, while the left gonad was preserved after removal of the cyst from the parenchyma, as requested by patient’s parents. This choice was justified by the possible residual functionality of the gonad, which had a macroscopically normal appearance. Therefore, despite the risks, it was decided, in agreement with the parents, not to remove it and to monitor it periodically through noninvasive radiological investigations.

The patient subsequently began a multidisciplinary follow-up with serial gynecological and pediatric endocrinological assessments (every six months). Gynecological examination revealed a female vulva, no perineal hair, and a blind-ended vagina with a total length of approximately 7 cm. During the last endocrinological evaluation (at 17 years), auxological parameters (height 167 cm, 79th centile, and weight 66 kg, 87th centile) were normal, with no lymphadenopathy, no axillary hair found, Tanner stage M4 bilateral (breast development with small areolas and depressed nipples) without concomitant hormonal replacement therapy, and nearly complete pubertal development. Hormonal tests were consistent with an advanced pubertal stage and testosterone levels were elevated for the female sex, without clinical manifestations of hyperandrogenism: estradiol 32.7 pg/mL (n.v. 20–40 pg/mL), testosterone 801 ng/dL (n.v. 300–1200 ng/dL), luteinizing hormone (LH) 23.7 mIU/mL (n.v. 1–30 mIU/mL), and follicle-stimulating hormone (FSH) 3.3 mIU/mL (n.v. 1–12 mIU/mL). Considering all these results and the normal female pubertal development, it was deemed unnecessary to start estrogen replacement therapy. The patient is currently undergoing regular follow-up at our institution with biannual check-ups.

## 3. Discussion and the Case-Based Review of the Medical Literature

CAIS patients have normal female external genitalia despite having a 46,XY karyotype and undescended testes: this is the result of the body being unresponsive to the signaling effects of testosterone, which is crucial for the masculinization process during fetal development. 

The presence of the SRY region in the fetal stage leads to the development of testes in the abdomen, but the androgen receptor mutation makes testosterone ineffective for regular male genitalia development [7,8]. Consequently, male external genitalia are absent. The internal female genitalia are also lacking because the testes produce anti-Müllerian hormone (AMH): AMH plays a significant role in the regression of Müllerian ducts during the embryonic stage, inhibiting the development of the uterus, cervix, and proximal vagina. However, the distal part of the vagina may be present, albeit shorter and blind-ended [9,10]. Puberty onset may be delayed and progress may be slower compared with healthy females. Either way, some patients with CAIS may still undergo normal female pubertal development with the onset of thelarche and female fat distribution; in fact, in CAIS patients, estradiol derives from the aromatization of testosterone, thus contributing to the development of secondary female sexual characteristics [1]. Conversely, pubic and axillary hair is sparse due to the ineffective action of androgens on hair follicles. Patients are often taller than the average due to the Y chromosome’s influence on height [10], suggesting that height is not totally androgen-dependent. 

During puberty, testosterone levels in males vary widely but generally stabilize between 300 and 1200 ng/dL, with estradiol levels around 20–40 pg/mL. In females, estradiol levels during puberty typically range between 20 and 350 pg/mL, with significant variations during the menstrual cycle, while testosterone levels are negligible [5]. As for the hormone levels of patients with CAIS, they typically show elevated LH and normal FSH, with testosterone levels falling within the normal male range but relatively elevated for a female and estradiol levels in the lower female range [11,12,13]. 

This hormonal pattern is due to a series of factors:-Lack of negative feedback on LH [14]: in patients with CAIS, early puberty LH levels are within the normal range for pubertal females or males, but they rise sharply during late puberty, similar to levels seen in mid-cycle females. This late but marked and continuous increase in LH is a sign that the testosterone receptor in the pituitary/hypothalamus is defective and unable to suppress LH release even though testosterone levels are extremely high.-Intact feedback from inhibin and other proteins [15]: inhibin, produced by Sertoli cells in the testes (which are present in patients with CAIS), is a protein hormone that acts by inhibiting follicle-stimulating hormone (FSH) and, more precisely, in women, it stimulates the maturation capacity of ovarian follicles (and is considered a marker of follicular reserve), whereas in men, it indirectly affects gamete development and controls spermatogenesis with a feedback mechanism on FSH secretion. Since these proteins specifically regulate FSH and are not significantly altered by the presence of androgens or their insensitivity, FSH levels tend to remain normal. Furthermore, FSH inhibition is less dependent on androgen levels than LH, thus explaining why FSH remains normal even when LH is elevated.-Aromatization of testosterone [16]: a portion of the testosterone produced by the gonads in patients with CAIS is converted into estradiol via the aromatase enzyme. This leads to estradiol levels within the lower female range. However, since the amount of testosterone converted is limited, estradiol levels do not reach the typical levels observed in women of reproductive age but fall within the lower range of the female spectrum.

Diagnosis of CAIS is usually suspected based on the clinical presentation and symptoms, such as primary amenorrhea; then, it becomes crucial to confirm the clinical suspicion by karyotype analysis, which will show a 46,XY result. This confirmation is critical as it distinguishes CAIS from other conditions with similar presentations. The definitive diagnosis requires AR genotyping. Diagnostic imaging, such as pelvic ultrasound or MRI, usually confirms the absence of Müllerian structures and identifies testes. In addition, a thorough differential diagnosis should include Müllerian agenesis, also known as Mayer–Rokitansky–Küster–Hauser syndrome [17], characterized by an absent uterus and a blind vaginal pouch. Müllerian agenesis presents with a distinct set of clinical features compared with CAIS, making differential diagnosis essential for appropriate management. Indeed, individuals with Müllerian agenesis have a 46,XX karyotype with normal ovarian function and normal concentrations of androgens and estrogens, but they may present with primary amenorrhea as the first manifestation of the condition, similar to what occurs in patients with CAIS.

The necessity of gonadectomy in individuals with CAIS continues to be a major topic of debate. CAIS is associated with an increased risk of testicular germ cell tumors (TGCT), which supports the removal of gonads to prevent testicular cancer. On the other hand, postponing gonadectomy until at least puberty allows for natural pubertal development, thanks to estradiol production from the peripheral aromatization of retained testicular testosterone. Delaying the surgery enables the individual to benefit from the physiological effects of puberty, which can be crucial for their overall development and well-being.

Recent studies have further elucidated this debate, demonstrating that the risk of developing TGCT is a significant factor in deciding the timing of gonadectomy. A recent review [4] analyzed 62 publications on the incidence of gonadal neoplasms in adult women diagnosed with CAIS, estimating an oncological risk of approximately 14% during adulthood. However, the appropriate timing for performing gonadectomy remains a debated issue to date. A consensus statement on the management of intersex disorders [18] (commonly known as ‘disorders of sex development’) has analyzed the diverse clinical and surgical needs of patients with CAIS. Experts have suggested delaying gonadectomy in CAIS as consensus has not established recommendations regarding the optimal age for this procedure, despite affirmations that testes should be removed in patients with complete or partial androgen insensitivity to prevent malignancy in adulthood [19]. The absence of a definitive consensus highlights the importance of individualized care for each patient. In a recent article, Hughes and Deeb suggested that gonadectomy should be deferred until after puberty. This approach would allow for the natural progression of puberty, giving patients the opportunity to make informed and independent decisions regarding the management of their condition [19], although preserving one of the gonads does not guarantee complete autonomy in estrogen production.

Testicular germ cell tumors (TGCT) constitute roughly 1–1.5% of all tumors in males and are the most common malignant cancer affecting individuals between the ages of 15 and 40. While TGCT incidence is markedly lower in childhood and adolescence, it can exceed 22% in adulthood [3]. The latest WHO classification indicates that most TGCTs arise from noninvasive lesions termed germ cell neoplasia in situ (GCNIS) and pre-GCNIS [3,4,5,6], which represent a precancerous state that has the potential to evolve into invasive cancer if not monitored and managed appropriately. 

CAIS could be associated with increased TGCT incidence compared with the general population; seminoma and gonadoblastoma are the most common histological variants, alongside other malignant tumors, such as choriocarcinomas, teratomas, and various other rarer forms of tumors [3].

Estimating cancer incidence in CAIS patients is challenging due to evolving management practices. Nonetheless, the medical literature indicates a general 5% risk in androgen insensitivity syndrome, with a prevalence of <1% in CAIS [20,21]. Moreover, the risk of malignant progression rises with age, remaining rare in the prepubertal stage, unlike other DSDs, such as partial androgen insensitivity syndrome. In contrast to the general population, while GCNIS often progresses to invasive cancer, malignancy in CAIS patients typically manifests as pre-GCNIS or GCNIS with low invasive potential, usually occurring in late adulthood. Multiple surveillance options are available to assess the risk of malignant transformation if gonadectomy is delayed [22,23,24]. Imaging studies are pivotal in the assessment of undescended gonads, particularly in CAIS, and MRI has emerged as a valuable tool to discern histopathological features in such patients. MRI provides detailed imaging that can help in distinguishing between benign and malignant conditions. A study conducted by Nakhal et al. [25] used MRI prior to gonadectomy and correlated imaging findings with histopathological outcomes. Common abdominal or pelvic MRI observations included para-testicular cysts and Sertoli cell adenomas, both indicative of benign diseases, a correlation reaffirmed by histopathological examination with statistically significant agreement. The use of MRI in this context has proven effective in identifying benign conditions, which can reduce the immediate need for surgical intervention. Importantly, no issues of invasive cancer were identified. Consequently, the study advocates for maintaining post-puberty gonadectomy as the standard of care, given the inability of MRI to detect microscopic premalignant lesions.

Laparoscopic gonadopexy, followed by ultrasound imaging, might serve as an alternative for those patients deferring gonadectomy; indeed, preserving one or both gonads reduces the risk of insufficient ovarian estrogen production although it is not guaranteed that the residual gonadal tissue will be sufficient to produce adequate levels of estrogen. In fact, ultrasound assessment has proven beneficial in monitoring gonads in CAIS patients after this surgical procedure [26]: it was performed in 6 patients (ages 3 to 20) to stabilize the gonads in a fixed position near the anterior abdominal wall. Ultrasound remains a key tool for post-surgical monitoring, providing valuable information on the positioning and condition of the gonads. Interestingly, the gonads could not be reliably detected before surgery, but post-operative ultrasound did find their exact position. Gonadal biopsies post-gonadopexy revealed no evidence of germ cell tumors, confirming the overall efficacy of this approach and supporting the continued use of gonadopexy as a viable management option.

Despite the usefulness of pelvic ultrasound in studying gonads, it is operator-dependent; moreover, in pediatric patients, if not performed by a dedicated resident, it may fail to identify pelvic-located gonads [27]. Therefore, the oncological risk in patients with CAIS justifies the performance of second-level exams such as MRI. Immunohistochemical markers have been studied to stratify the risk of germ cell tumors in CAIS, especially seminomas, the most common germ cell tumor type in these patients [3]. The use of immunohistochemical markers provides additional layers of diagnostic detail that can aid in assessing cancer risk. Blood tumor markers such as human chorionic gonadotropin (hCG), lactate dehydrogenase (LDH), and alpha-fetoprotein (AFP) have been investigated, with LDH showing elevated levels in a significant proportion of cases. Elevated levels of LDH can be indicative of seminoma presence, making it a useful marker for monitoring. Specifically, LDH levels are elevated in 40–60% of patients with seminoma [28]. Currently, it is recommended to measure serum concentrations of hCG and/or LDH before and after orchidectomy in patients with seminomas [18,19,20,21,28]. These markers represent a noninvasive option to help in the diagnosis of germ cell tumors, contributing to a comprehensive monitoring strategy. OCT3/4, short for Octamer-binding transcription factor 3/4, is a transcription factor involved in the regulation of gene expression during embryonic development and in the maintenance of pluripotent stem cells: gonadal biopsy during puberty with immunohistochemical staining for OCT3/4, which is a marker of germ cell tumors, has been proposed to help stratify the oncological risk [18,20]. 

Gonadectomy has been recommended for patients in the higher-risk group, which included patients with gonadal dysgenesis (XY), partial androgen insensitivity, and Denys–Drash syndrome [28]. 

Hormonal replacement therapy (HRT) becomes imperative following bilateral gonadectomy to mitigate hypoestrogenism [29,30,31,32]. The timing and dosage of HRT are tailored to individual needs, reflecting the complexity of managing hormonal changes post-gonadectomy. Traditionally, HRT for CAIS patients has involved the use of estrogen therapy, yet determining the optimal daily dosage is elusive. Current practices involve careful adjustment of estrogen levels to match physiological requirements and individual patient needs. Consequently, initiating HRT at the lowest feasible dose, like oral ethinyl-estradiol at 2.5–5 µg/day or 50–100 ng/kg/day, is recommended, with subsequent gradual escalation toward adult dosages (20–25 µg/day) [32] to emulate the physiological hormone secretion pattern. For prepubertal individuals, HRT should be incrementally adjusted every six months to drive complete feminization, encompassing milestones like breast development, alterations in body composition, and attainment of female physique, which is typically achieved over a two-year period. This approach allows for controlled and gradual feminization, which can be crucial for psychological and physical development. Upon achieving full breast development, therapy should be continued at a regular daily dose [29,33,34]. Continued therapy ensures the maintenance of secondary sexual characteristics and hormonal balance.

A critical aspect of the gonadectomy decision is the psychosocial impact on the individual. This aspect cannot be overstated as the psychological and social dimensions play a significant role in the overall treatment strategy. Managing CAIS requires a specific approach that considers the emotional and psychological challenges associated with this condition. Psychological counseling and social support are crucial in this context. Effective psychological support can help patients navigate the complexities of their condition and make informed decisions about their treatment options. The decision to undergo gonadectomy carries risks and benefits. The procedure can mitigate the risk of developing gonadal tumors and alleviate complications related to androgen insensitivity; however, the removal of gonads may impact sexual health, gender identity, and overall self-esteem. This complex decision underscores the necessity of a personalized approach that considers both medical and emotional aspects. It is reasonable to advocate for a conservative management strategy for CAIS starting in adolescence (i.e., around ages 14 to 15) [35] (Figure 1).

Regarding the genetic investigation conducted on our patient, we identified the hemizygous variant for the c.159_207del mutation (p.Leu54Serfs*105). This mutation is not described in the medical literature as a pathogenic cause of CAIS; however, it has been considered pathogenic as it induces the formation of a premature stop codon, leading to altered production of the androgen receptor. The AR protein consists of 919 amino acid residues and is composed of four functional domains, while more than 600 different mutations might be causative of the syndrome [36]: the identification of the underlying mutation is essential for the diagnosis of CAIS as hormone levels, although they may suggest this condition in the presence of a DSD, do not allow for a definitive diagnosis [37]. Jing He et al. [38] analyzed six patients with a genetic diagnosis of CAIS, finding in 4 patients a mutation different from those described in the literature and highlighting that, regardless of the type of genetic alteration, AR dysfunction leads to DSD and androgen insensitivity. The same authors did not find any correlation between the patients’ baseline hormone levels and the type of mutation identified. It is debated whether the different mutations can influence the phenotypic characteristics of the patient and the subsequent oncological risk [39]. It is commonly believed that different mutations can cause similar clinical phenotypes by equally leading to AR dysfunction, although missense mutations appear to be the most frequent in CAIS, while other alterations cause DSD with different characteristics [36]. There are also AR mutations giving rise to male phenotypes, even with preserved fertility [14,15,16,40]. All these findings highlight the complexity of the clinical picture of CAIS and uphold the necessity of a multidisciplinary management.

## 4. Future Directions

Research on DSD and CAIS is continually advancing, underscoring the need for ongoing investigations and more specific diagnostic methods. Future efforts should concentrate on broadening genetic testing to discover new mutations and comprehend their effects on phenotypic diversity and cancer risk. New research studies should be warranted to provide new insights into the mutations associated with AIS; in particular, the focus should be on creating personalized medical treatment plans that include endocrinologists, geneticists, psychologists, and surgeons so that all aspects of the patient’s health and well-being are addressed comprehensively. 

The integration of emerging research findings into clinical protocols will be crucial for optimizing medical care and improving long-term outcomes for CAIS patients.

## 5. Conclusions

Our case-based review highlights the importance of a comprehensive approach to diagnosing patients with CAIS and illustrates the complexity of their general management. The risks of gonadectomy and subsequent HRT in CAIS should be discussed with patients and their families. Despite significant improvements in laparoscopic techniques, surgery bears intrinsic risks, and furthermore, gonadectomy necessitates lifelong hormonal replacement treatment. Psychological implications, such as gender identification or depression, should also be taken into consideration in the overall evaluation of the patient. The low neoplastic risk before puberty supports a conservative approach, allowing for spontaneous pubertal development, although the preservation of one or both gonads does not exclude the need for hormone replacement therapy for the development and progression of secondary sexual characteristics. To this end, periodic check-ups through imaging studies such as MRI and gonadal ultrasound, may, however, not reveal precancerous lesions. Laparoscopic gonadopexy, followed by ultrasound imaging, might serve as an alternative for those patients deferring gonadectomy, facilitating the visualization of fixed-position gonads. Nonetheless, this approach involves invasive surgery, which may not be deemed appropriate by certain patients. Although tumor markers like hCG and LDH show elevated levels in seminomas, there are no current guidelines recommending their use in CAIS.

## Figures and Tables

**Figure 1 diseases-12-00235-f001:**
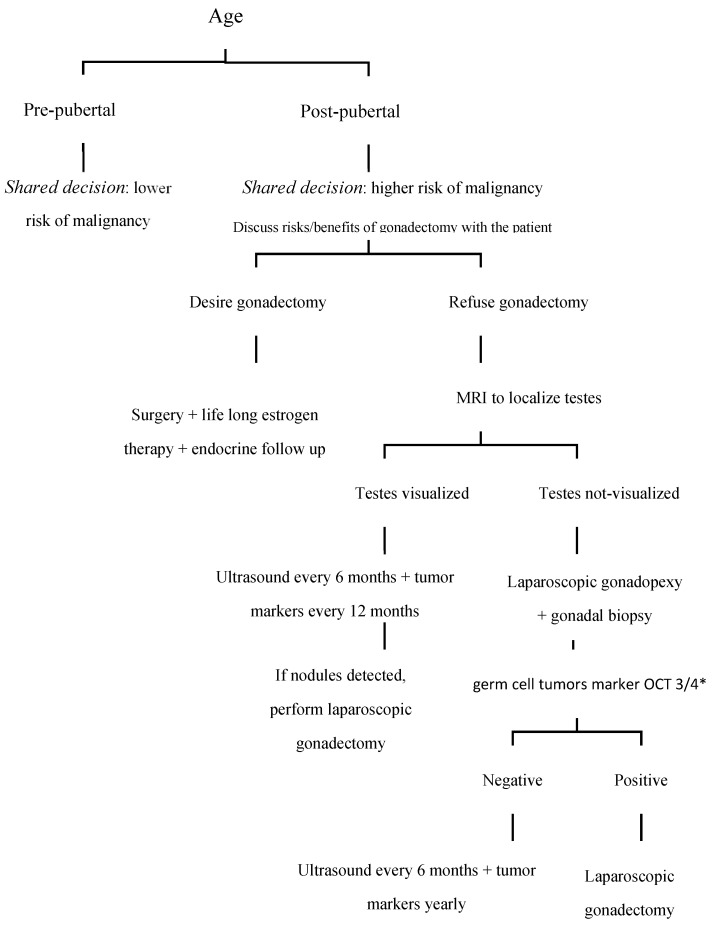
Proposal for an algorithm for the management of patients with complete androgen insensitivity syndrome. *** OCT3/4:** Octamer-binding transcription factor 3/4.

## Data Availability

The data presented in this study are available on request from the corresponding author. The data are not publicly available due to privacy.

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
