# Peer review of "Complete Androgen Insensitivity Syndrome in a Young Girl with Primary Amenorrhea and Suspected Delayed Puberty: A Case-Based Review of Clinical Management, Surgical Follow-Up, and Oncological Risk"

_diseases, 2024, doi:10.3390/diseases12100235_

Round 1
Reviewer 1 Report
Comments and Suggestions for Authors
In the uploaded word document

Author Response
Reviewer 1
This clinical case based review describes a patient with complete androgen insensitivity syndrome, followed by a narrative review of CAIS management.
The case report itself is interesting, especially because it reveals a new variant, not described in the medical literature. The review highlights the importance of the oncological risk and careful and shared decision making with patient and her family. Overall, the introduction and the section of the case presentation are well written; concise but still informative.
However, I do have some minor remarks and some major concerns.
Dear Reviewer,
we appreciate your feedback on our manuscript. We have made several modifications to the article based on your comments and those of the other reviewer. Below you will find a point-by-point response to your observations.
- In general, the part on gonadectomy (from line 168 onwards, to line 300) is not well written. It is very repetitive (some topics are addressed multiple times, information is mentioned twice, ...), does not have a logical order. I would strongly recommend rewriting for more conciseness, better structure and buildup.
Thank you for your comment. We have revised the paragraph by adding more information and attempting to clarify the concepts.
- I strongly advise not to integrate figure 1 in the manuscript. The level of evidence is low, it is rather a suggestion of the authors. It also contains information that is not integrated in the text (DEXA).
Thank you for your observation. Figure 1 of our manuscript was included in the text as it was explicitly required for the submission of our manuscript. We have revised the figure to clarify certain concepts; based on the Editor's decision, we may choose not to include it in the final version of the paper and retain it as supplementary material.
- Line 85-6: on the altered AR transcription: is this completely blocking the transcription of AR? Which percentage of AR is still produced? Is “complete” AIS based on clinical or molecular diagnosis? Please elaborate.
We have included additional information; the mutation found in our patient leads to the formation of a premature stop codon, resulting in the transcription of an abnormal and non-functional protein. The diagnosis was both clinical and molecular.
- Line 91-92: Please elaborate on the reasoning of this choice of the parents, to preserve the left gonad.
This choice was justified by the possible residual functionality of the gonad, which had a macroscopically normal appearance. Therefore, despite the risks, it was decided, in agreement with the parents, not to remove it and to monitor it periodically through non-invasive radiological investigations.
- Line 99: please add that the this thelarche was present without HRT
- Line 136-139: why is estradiol no longer playing a role in the negative feedback system?
- Line 140: ass the previous paragraph is on the lack of negative feedback of LH, I would clarify this sentence with “Intact” feedback from inhibin and other proteins.
- Line 166-167: I would specify that patients with Mullerian agenesis also present with primary amenorrhea?
- Line 181-187: Too long and unclear sentence: please rephrase.
- Lines 202-203: how is this different from lines 177-178?
- Lines 225: The reasoning behind gonadopexy is not explained.
- Lines 244-246: repetitive sentences
- Line 257-259: I do not understand why this information is added here.
- Line 283-293: repetitive
- Line 333: medical care?
We thank you once again for your comments, which have allowed us to improve the quality of our manuscript. We have revised the information and reworked the concepts in the paragraphs you indicated, as well as in the entire discussion section. Redundant concepts have also been removed. The changes made are highlighted in the manuscript.
- Line 343: lifelong HRT might also be necessary for patients without gonadectomy, as estradiol levels might not be high enough to protect bone quality and to maintain sexual characteristics. I do not find it reasonable to integrate this as an argument in clinical decision making of having gonadectomy or not (as it is possible that HRT is necessary in both cases).
Thank you for your observation; indeed, gonadectomy certainly makes hormone replacement therapy necessary for the development of secondary sexual characteristics, whereas preserving at least one gonad does not exclude the need for estrogen therapy. Although there is a possibility of not resorting to hormone therapy, we agree that this information is not decisive in the clinical management of patients. We have clarified this concept in the text and revised the paragraph
We hope that the changes made might meet your overall expectations.
Thank you so much again for your time and for your valuable advice aimed at improving our manuscript.
My personal kindest regards,
Dr Giorgio Sodero
Reviewer 2 Report
Comments and Suggestions for Authors
The Authors hereby describe a case report of CAIS and made an effort to review the literature. The case is interesting and well described, the literature review is complete.
I have just a few concerns:
- lines 47-49: please add the neonatal presentation with female external genitalia and inguinal palpable gonads that is not so rare
- lines 68-72: please move the lines to discussion
- Are AMH and Inhibin B levels available in your case report?
Author Response
Reviewer 2
The Authors hereby describe a case report of CAIS and made an effort to review the literature. The case is interesting and well described, the literature review is complete.
I have just a few concerns:
Dear Reviewer, thank you for your effort in reviewing our manuscript. We have made significant changes to the manuscript, incorporating the missing information requested by you and the other reviewer.
Below you will find a point-by-point response to your observations.
- lines 47-49: please add the neonatal presentation with female external genitalia and inguinal palpable gonads that is not so rare
Thank you for your comment. We have revised the paragraph based on your suggestions and those of the other reviewer.
- lines 68-72: please move the lines to discussion
Modified. Thank you for your comment.
- Are AMH and Inhibin B levels available in your case report?
We have incorporated this information into the manuscript.
We hope that the changes made might meet your overall expectations.
Thank you so much again for your time and for your valuable advice aimed at improving our manuscript.
My personal kindest regards,
Dr Giorgio Sodero